# Risk Stratification in Hypertrophic Cardiomyopathy. Insights from Genetic Analysis and Cardiopulmonary Exercise Testing

**DOI:** 10.3390/jcm9061636

**Published:** 2020-05-28

**Authors:** Damiano Magrì, Vittoria Mastromarino, Giovanna Gallo, Elisabetta Zachara, Federica Re, Piergiuseppe Agostoni, Dario Giordano, Speranza Rubattu, Maurizio Forte, Maria Cotugno, Maria Rosaria Torrisi, Simona Petrucci, Aldo Germani, Camilla Savio, Antonello Maruotti, Massimo Volpe, Camillo Autore, Maria Piane, Beatrice Musumeci

**Affiliations:** 1Department of Clinical and Molecular Medicine, Sapienza University, 00100 Rome, Italy; vittoriamastromarino@libero.it (V.M.); giovanna.gallo@uniroma1.it (G.G.); dariogiordano@outlook.com (D.G.); mara.torrisi@uniroma1.it (M.R.T.); simona.petrucci@uniroma1.it (S.P.); aldo.germani@uniroma1.it (A.G.); massimo.volpe@uniroma1.it (M.V.); camillo.autore@uniroma1.it (C.A.); maria.piane@uniroma1.it (M.P.); beatrice.musumeci@uniroma1.it (B.M.); 2Unit of Pediatric Cardiology and Cardiac Surgery, Sant’Orsola Hospital, 40100 Bologna, Italy; 3Cardiac Arrhythmia Center and Cardiomyopathies Unit, San Camillo—Forlanini Hospital, 00100 Rome, Italy; elisabettazachara@gmail.com (E.Z.); re.federica77@gmail.com (F.R.); 4Centro Cardiologico Monzino, IRCCS, 20100 Milan, Italy; piergiuseppe.agostoni@unimi.it; 5Department of Clinical Sciences and Community Health, University of Milan, 20100 Milan, Italy; 6IRCCS Neuromed, 86077 Pozzilli (IS), Italy; maurizio.forte@neuromed.it (M.F.); maria.cotugno@neuromed.it (M.C.); 7UOC Medical Genetics and Advanced Cell Diagnostics, S. Andrea University Hospital, 00100 Rome, Italy; camilla.savio@ospedalesantandrea.it; 8Department of Scienze Economiche, Politiche e delle Lingue Moderne—Libera Università SS Maria Assunta, 00100 Rome, Italy; a.maruotti@lumsa.it; 9Department of Mathematics, University of Bergen, 5052 Bergen, Norway; 10School of Computing, University of Portsmouth, Portsmouth PO1, UK

**Keywords:** hypertrophic cardiomyopathy, cardiopulmonary exercise test, genetic testing

## Abstract

The role of genetic testing over the clinical and functional variables, including data from the cardiopulmonary exercise test (CPET), in the hypertrophic cardiomyopathy (HCM) risk stratification remains unclear. A retrospective genotype–phenotype correlation was performed to analyze possible differences between patients with and without likely pathogenic/pathogenic (LP/P) variants. A total of 371 HCM patients were screened at least for the main sarcomeric genes *MYBPC3* (myosin binding protein C), *MYH7* (β-myosin heavy chain), *TNNI3* (cardiac troponin I) and *TNNT2* (cardiac troponin T): 203 patients had at least an LP/P variant, 23 patients had a unique variant of uncertain significance (VUS) and 145 did not show any LP/P variant or VUS. During a median 5.4 years follow-up, 51 and 14 patients developed heart failure (HF) and sudden cardiac death (SCD) or SCD-equivalents events, respectively. The LP/P variant was associated with a more aggressive HCM phenotype. However, left atrial diameter (LAd), circulatory power (peak oxygen uptake*peak systolic blood pressure, CP%) and ventilatory efficiency (C-index = 0.839) were the only independent predictors of HF whereas only LAd and CP% were predictors of the SCD end-point (C-index = 0.738). The present study reaffirms the pivotal role of the clinical variables and, particularly of those CPET-derived, in the HCM risk stratification.

## 1. Introduction

Hypertrophic cardiomyopathy (HCM), the most common genetic heart disease, inherited with an autosomal dominant pattern, incomplete penetrance and variable expressivity, is characterized by markedly different instrumental and clinical spectra [1,2,3]. Accordingly, there is always great interest in investigating approaches potentially able to identify early those HCM patients at high risk of cardiovascular events both in terms of sudden cardiac death (SCD) and heart failure (HF). Indeed, albeit the SCD is a devastating but relatively rare event, HF development and its related complications still represent an incoming concern in HCM patients [4,5,6].

In such a context, due to its genetic nature, many researchers attempted not only an HCM genotype–phenotype correlation but also a possible genotype-based risk stratification. Indeed, HCM is predominantly a sarcomeric disease and variants in *MYH7* and *MYBPC3* genes, encoding for the cardiac thick myofilament proteins β-myosin heavy chain and myosin binding protein C, respectively, together account about 50% of the HCM families. Conversely, likely pathogenic/pathogenic (LP/P) variants in thin filament protein genes, such as *TNNT2*, *TNNI3* and *TPM1* encoding for cardiac troponin T, troponin I and alpha-tropomyosin, account for less than 10% [6,7,8,9]. Several studies have demonstrated that being carriers of sarcomeric variants might exert a negative prognostic impact on outcome, as well as multiple simultaneous variants, such as the so-called “gene dosage effect” [8,9,10,11,12,13]. What remains unclear, up to now, is whether the genetic profile of the single HCM patient might provide a real significant incremental risk prediction beyond the clinical risk factors, including those derived from a maximal cardiopulmonary exercise test (CPET) [2,3]. Indeed, growing evidence suggests that a CPET assessment, combined with other clinical and instrumental variables, represents a useful tool in stratifying both the SCD and the HF-related events’ risk in HCM patients [14,15,16,17,18].

Therefore, the current multicenter retrospective study investigates a possible adjunctive role of genetic testing analysis in the HCM patients’ management over the main clinical and functional parameters. Particularly, a genotype–phenotype correlation was performed to analyze possible differences between HCM patients with and without LP/P variants with respect to their main clinical and functional features and, mainly, their SCD and HF-related events’ rate.

## 2. Methods

### 2.1. Study Sample

Data from a total of 665 consecutive outpatients with HCM were analyzed. All patients were part of a cohort recruited and prospectively followed in three HCM Italian centers between September 2007 and December 2019: Azienda Ospedaliera Universitaria Sant’Andrea, “Sapienza” University, Rome (*n* = 437); Azienda Ospedaliera San Camillo Forlanini, Rome (*n* = 189); Centro Cardiologico Monzino, University of Milan, Milan (*n* = 39). The diagnosis of HCM was based on maximal wall thickness (MWT) ≥ 15 mm unexplained by abnormal loading conditions or in accordance with published criteria for the diagnosis of disease in relatives of patients with unequivocal disease [2,3]. Patients with known metabolic diseases or syndromic causes of HCM were excluded from the present study.

The study complied with the ethical standards of the Declaration of Helsinki and was reviewed and approved by the institutional ethics committees. Written informed consent was obtained from all participants. The authors from each participating center guarantee the integrity of data from their institution and have agreed to the article as written.

### 2.2. Patients Clinical and Functional Assessment

Data were independently collected at each participating center using a uniform methodology. Each HCM patient underwent a clinical assessment, including history with pedigree analysis and the New York Heart Association (NYHA) classification, 24 h ECG Holter monitoring, transthoracic Doppler echocardiography and maximal CPET. The usual five SCD risk factors were also collected [2,3]: (a) FH-SCD (history of HCM-related SCD in at least one first-degree or other relatives younger than 50 years old); (b) massive left ventricular (LV) hypertrophy (maximal wall thickness, MWT, ≥30 mm); (c) at least one run of nonsustained ventricular tachycardia (NSVT, ≥3 consecutive ventricular beats at a rate of ≥120 beats per minute and <30 s in duration on 24 h ECG Holter monitoring); (d) unexplained syncope judged inconsistent with neurocardiogenic origin; (e) abnormal blood pressure response to exercise (ABPRE, failure to increase systolic blood pressure, SBP, by at least 20 mmHg from rest to peak exercise or a fall of ≥20 mmHg from SBP).

The following echocardiographic measurements, obtained according to the international guidelines [19], were considered: LV end-diastolic diameter (LVEDd, parasternal long axis), the greatest LV thickness (MWT, measured at any LV site), left atrial diameter (LAd, parasternal long axis), the highest maximal LV outflow tract gradient among those measured at rest, in orthostatic position and after Valsalva maneuver (LVOTGmax, apical four-chamber view) [20] and LV ejection fraction with Simpson’s biplane methods (LVEF, apical four-chamber view).

All CPETs were performed using an electronically braked cycle ergometer. A personalized ramp exercise protocol was performed, aiming at a test duration of 10 ± 2 min [21]. The exercise was preceded by a few minutes of resting breath-by-breath gas exchange monitoring and by an unloaded warm-up. In the absence of clinical events, CPET was interrupted when patients stated that they had reached maximal effort. A 12-lead ECG, diastolic and systolic blood pressure were recorded during CPET, in order to obtain the following parameters: rest heart rate (HR), peak HR, %pHR ((peak HR/(220 − age)) × 100), and ΔSBP (peak SBP − rest SBP) [22]. A breath-by-breath analysis of expiratory gases and ventilation (VE) has been performed, and peak values were obtained in the last 20 s of exercise. The predicted peak VO_2_ was determined by using the gender-, age- and weight-adjusted formula. Circulatory power (CP = peak VO_2_ × SBP) was obtained considering peak VO_2_ value as a percentage of predicted (CP%) [19,23]. Anaerobic threshold (AT) was measured by V-slope analysis of VO_2_ and VCO_2_, and it was confirmed by ventilator equivalents and end-tidal pressures of CO_2_ and O_2_ [24]. The end of the isocapnic buffering period was identified when VE/VCO_2_ increased and the end-tidal pressure of CO_2_ decreased. VE/VCO_2_ slope was calculated as the slope of the linear relationship between VE and VCO_2_ from the 1st minute after the beginning of the loaded exercise and the end of the isocapnic buffering period [24].

### 2.3. Genetic Testing

All patients included in the study received genetic counseling and underwent genetic testing for HCM, performed by Sanger sequencing (from 2007 to 2010) or NGS (next-generation sequencing, from 2011 to 2019). The genes analyzed in the participating laboratories from the HCM centers had changed and/or added over time. In this retrospective study, we report the genotype–phenotype correlation analysis only for those patients screened at least for the sarcomeric genes *MYBPC3* (myosin binding protein C, cardiac) *MYH7* (β-myosin heavy chain, cardiac), *TNNI3* (troponin I, cardiac) and *TNNT2* (troponin T type 2, cardiac). All coding regions and boundaries of flanking introns ± 25 were analyzed for all genes considered in this study. In addition, all variants reported in this paper, identified by NGS technology, were validated by Sanger sequencing. Patients with variants located in nonsarcomeric genes were also excluded.

All the identified variants were re-evaluated based on new evidence from the scientific literature and classified according to the criteria of the American College of Medical Genetics and Genomics (ACMG) [25,26]. Only the genetic variants predicted to alter the protein and with a minor allele frequency (MAF) ≤0.2% (considering the prevalence of HCM disease in the general population), were considered. For this evaluation, we used MAF data derived from GnomAD (Genome Aggregation Database https://gnomad.broadinstitute.org/). The clinical classification of variants was carried out according to the 5-class system: benign (B), likely benign (LB), likely pathogenic (LP), pathogenic (P) and variants of uncertain significance (VUS). Genetic results were considered informative for LP or P variants, noninformative for B, LB or VUS variants. Variants are reported using the Human Genome Variation Society nomenclature guidelines (https://varnomen.hgvs.org/).

### 2.4. Clinical Outcomes

All patients had planned clinical reviews every 6–12 months or earlier according to the clinical status. Follow-up duration was defined as the time interval between the clinical examination and either the first event or the last visit/telephone interview in case of no events.

The HF end-point was represented by the following events: death due to HF, cardiac transplantation, progression to a stable NYHA class III–IV due to an end-stage phase with or without LVEF < 50% (hypokinetic dilated phase or restrictive phenotype evolution), severe functional deterioration leading to hospitalization for septal reduction, hospitalization due to HF symptoms or signs development. The SCD end-point was also tested, which included SCD or an equivalent event. SCD was defined as witnessing sudden death with or without documented ventricular fibrillation or death within 1 h of new symptoms or nocturnal deaths with no antecedent history of worsening symptoms. Aborted SCD during follow-up and appropriate ICD therapies (defined as intervention triggered by ventricular fibrillation or rapid ventricular tachycardia at >180 bpm) were considered equivalent to SCD in accordance with previous studies [17,18,27,28].

The causes of death, as well as the other events, were ascertained by experienced cardiologists at each center using hospital and primary health care records, death certificates, post-mortem reports, and interviews with relatives and/or physicians. To avoid a composite cardiovascular end-point including also cerebrovascular events, due to the small number of such events and in accordance with other studies by our research group [17,18,22], death due to ischemic or hemorrhagic stroke (n. 2 events) and nonfatal cerebrovascular (n. 5 events), as well as death due to noncardiovascular causes (n. 2 events), were excluded from the survival analysis.

### 2.5. Statistical Analysis

All data are expressed as mean ± standard deviation or as absolute number (percentage). Preliminarily, an extension of the Shapiro–Wilk test of normality was performed. Categorical variables were compared with a difference between proportion tests whereas a two-sample *t*-test was used to compare the continuous data between the two study groups (no variants and VUS Versus LP/P variants). In comparing the two populations, the variance was estimated separately for both groups and the Welch–Satterthwaite modification to the degrees of freedom was used.

We, therefore, focused on the distribution of survival times by adopting the Cox proportional hazards regression model. We performed a backward selection of the predictors to be included in the model. A 5% significance level was used in the backward elimination procedure to select covariates for the final multivariate model for the combined as well as the HF end-point while, due to the low events’ number, a 15% significance level was adopted for the SCD one. To avoid the inclusion of collinear variables in the multivariate Cox analysis, we built several models in which VO_2_-derived variables, known to be collinear, were added to the prognostic model one at a time. We retain the model with the best trade-off between model complexity and model fit judged by the log-likelihood. We also performed a calibration analysis. We computed the average calibration error for both approaches and tested the observed versus average predicted probabilities for each class of risk. The Brier quadratic error score and a χ^2^ test of goodness of fit based on the Brier score were also checked. We did not find any clear indication of overfitting from the post hoc analysis and, consequently, in the present paper we simply reported results based on the backward elimination procedure only. Discrimination of variables included in the final multivariate model specification was performed by Harrell’s C-index. Therefore, we investigated the proportional hazards assumption by tests and graphical diagnostics based on scaled Schoenfeld residuals. Test of proportional hazards assumption for each covariate was obtained by correlating the corresponding set of scaled Schoenfeld residuals with the Kaplan–Meier estimate of the survival distribution. To check for the presence of influential observations, we produced a matrix of estimated changes in the regression coefficients upon deleting each observation in turn and comparing the magnitudes of the largest values to the regression coefficients.

Statistical analysis was performed using R (R Development Core Team, 2014). A *p*-value lower than or equal to 0.05 was generally considered as statistically significant.

## 3. Results

From an initial study sample of 665 consecutive HCM outpatients, a total of 294 patients (44%) were excluded because they did not undergo genetic testing (*n* = 197), because they were lost at follow-up (*n* = 39), because of the presence of nonsarcomeric variants (*n* = 22) or, eventually, because the genetic analysis was not performed according to the previously described inclusion criteria (*n* = 36). Thus, a total of 371 HCM patients were effectively enrolled and analyzed in the present study. The diagram displayed in Figure 1 resumes the step-by-step classification of the study population.

### 3.1. Genetic Results

Two hundred and three (55%) genetic tests were informative as they detected at least an LP/P variant, whereas the percentage of patients with VUS was 6% (*n* = 23 patients); 39% (*n* = 145 patients) did not show any P/LP variant or VUS (Figure 2). Excluding the B/LB variants, 124 unique variants were identified and detailed extensively in the Appendix A. These variants included 91 (73%) missense, 2 (2%) intronic, 10 (8%) frameshift, 8 (6%) splicing, 11 (9%) nonsense, 1 (1%) inframe variants (Appendix A). According to ACMG criteria, 94 variants were classified as LP/P and 30 of them as VUS. *MYBPC3* and *MYH7* resulted in the most mutated genes with 88 LP/P variants (75%, Figure 2). Twenty-four patients resulted to be carriers of multiple variants, considering those with at least one P/LP variant and other P/LP or VUS variants (Appendix A). Among these, 14 patients were double heterozygous with one variant in two different genes while the remaining 10 had multiple variants in the same gene. It was not possible to determinate the phase of these variants, since segregation studies among relatives could not be performed.

### 3.2. Clinical and Functional Characteristics

The demographic and clinical data of the entire cohort are reported in Table 1. The population mainly consisted of middle-aged predominantly male (64%) patients with a quite preserved NYHA class (NYHA I–II 94%). At the study run-in, echocardiographic evidence of the end-stage phase was present in 4%, atrial fibrillation in 3% and a septal myectomy had been performed in 11% patients. Documented cardiovascular comorbidities included systemic hypertension (27%), diabetes (6%) and coronary artery disease (4%).

Table 2 shows the comparison of clinical features between patients with and without LP/P variants. The LP/P variants group showed a younger age, a slightly higher prevalence of FH-SCD and ABPRE and a worse functional capacity in terms of pVO_2_, CP% and VE/VCO_2_ slope. With respect to the other clinical features, at the study run-in, the LP/P variants group had a greater prevalence of patients with end-stage phase (6% vs. 2%, *p* = 0.011) and atrial fibrillation (3% vs. 1%, *p* = 0.032), whereas no difference in the prevalence of previous myectomy was found (5% for both groups). Concerning the documented cardiovascular comorbidities, no difference was found in coronary artery disease (3% vs. 5%), whereas the LP/P variants group showed a lower prevalence of systemic hypertension (15% vs. 31%, *p* < 0.001) and diabetes (3% vs. 8%, *p* = 0.007).

### 3.3. End-Point Analysis

Median follow-up was 5.4 years (25th–75th centile: 2.3 to 8.1 years) with a total of 2271 patients-year. During the entire follow-up, a total of 129 (35%) patients experienced at least one of the pre-specified events. In patients who developed multiple events, time to the first was used as an event time cutoff and, accordingly, SCD or HF-related events at five-years’ cumulative hazard equal to 0.369 was estimated. Patients who completed the follow-up period before the tenth year were censored at the time of the last clinical evaluation.

A total of 14 SCD or SCD-equivalents were analyzed. Specifically, SCD occurred in three patients; four patients experienced a resuscitated SCD and seven patients had an appropriate ICD intervention. A total of 52 HF-related events were analyzed. Specifically, HF-related death occurred in two patients, seven patients underwent cardiac transplantation, 20 patients were hospitalized due to HF signs/symptoms, 12 patients were hospitalized for septal reduction procedure due to significant HF signs/symptoms development and 11 patients evolved to end-stage or restrictive phenotype evolution. Table 3 reports the detailed Cox proportional univariate survival analysis for both the study end-points. Most of the single variables were significantly associated with the HF end-point whereas few of them to the SCD end-point. Particularly, besides a number of clinical variables, the LP/P variant presence was significantly associated with the HF but not to the SCD end-point (Figure 3). Instead, at multivariate analysis, covariates showing significant effects for the primary HF-related end-point were the following: LAd, CP% and VE/VCO_2_ slope (C-index 0.839, *p* < 0.001) while LAd and the CP% remained independently associated with the SCD end-point (C-index 0.738, Table 4).

## 4. Discussion

The present multicenter retrospective study, conducted on a suitable cohort of consecutive HCM outpatients regularly followed at three Italian tertiary HCM centers, shows that HCM patients with LP/P sarcomeric variants tend to present a more aggressive form of disease with an earlier onset, a worse functional status and a greater risk of HF development and HF-related complications, compared to HCM patients with VUS or any variants. However, contextually, our data do not support a strict role of genetic testing over HCM patients’ comprehensive clinical assessment. Indeed, the LP/P variants presence does not emerge at multivariate analysis as an independent risk factor, being other instrumental variables (i.e., LAd, CP% and VE/VCO_2_ slope) much stronger outcome predictors.

Over the last decades a number of clinical features were investigated in order to identify those HCM patients at high risk of adverse events, both arrhythmic and HF-related. Indeed, albeit most of the cases show a benign course with a life expectancy equal to the general population [1], the SCD remains a rare but devastating event which is still the leading cause of death in the young population and athletes [2,3,6]. Furthermore, proper due to the improvement in the HCM management, both pharmacological and nonpharmacological (i.e., ICD, myectomy, LVAD/heart transplantation), there is a growing percentage of HCM patients who develop HF as well as HF-related complications [1,5,29]. The present study, as an ancillary result, confirms the abovementioned concern by showing a significant rate of HF-related events (*n* = 114 events) with a relatively low number of SCD and SCD-equivalent (*n* = 14 events) at a midterm follow-up.

Given the wide HCM clinical spectrum, its genetic nature and the need for targeted prevention strategies, previous studies sought to investigate and weigh the influence of sarcomere variants on the HCM clinical phenotype and outcome [7,8,9,10,11,12,13,16]. Indeed, HCM represents a sarcomeric disease with LP/P variants in *MYBPC3* and *MYH7* genes (thick filaments) together accounting for about 50% of the HCM families whereas LP/P variants in *TNNT2*, *TNNI3* (thin filaments) accounting for less than 10% [1,6]. In such a context, an old study by Olivotto and colleagues, conducted in the pre-NGS scenario on a cohort of 203 HCM patients with a median follow-up of 4.5 years, suggested that HCM patients with sarcomere variants had a greater probability of a worse outcome (i.e., cardiac death, nonfatal stroke, end-stage progression) compared to those with nonsarcomeric variants [8]. A more recent study by Li and colleagues, on a sample of 558 HCM patients with a median follow-up of 4.5 years, demonstrated that LP/P variants were associated to an early disease onset, to a high burden of established risk factors and, mainly, to a composite HF end-points (i.e., HF-related hospitalization, heart transplantation, HF-related death, progression to an end-stage phase) [10]. Further support to a possible role of genetic testing in the HCM risk stratification comes from Velzen and colleagues that, on a population of 626 HCM patients with a long-term follow-up (>10 years), confirmed that patients with LP/P variants had a more aggressive phenotype (i.e., young age, high prevalence of SCD risk factors) and found an independent association with all-cause mortality, HF-related and SCD mortality [12]. Eventually, the recent SHARE study, conducted on a large cohort of 2763 HCM patients followed-up for a median of 2.9 years, confirmed that patients carrying sarcomeric LP/P variants had an earlier disease onset and a greater risk of developing the overall composite outcome (SCD or SCD-equivalent, LVAD/cardiac transplantation, progression to an end-stage phase, all-cause mortality, atrial fibrillation and stroke) [13]. The present study confirms that HCM patients carrying LP/P variants have a worse clinical feature in terms of disease onset as well as of historical risk factors [29]. However, we proved just a univariate association between the HCM-mutated status and a composite HF end-point. Although a close comparison with the abovementioned studies remains difficult because of different methodological approaches (i.e., number of genes screened, variants classification, end-points’ construction), the most likely reason underlying this negative datum could be the optimal clinical and functional characterization of our study cohort. Indeed, we specifically challenged the prognostic impact of LP/P variants not only with the historical variables but also, and specifically, with the CPET-derived parameters. Thus, we showed originally that patients with LP/P variants, although younger than the counterpart, exhibited also a more severe functional limitation both in terms of pVO_2_ and ventilatory efficiency values. Furthermore, due to a concomitant blunted increase of SBP during exercise, HCM patients with LP/P variants showed significantly lower values of CP%. Each of the three abovementioned CPET-derived variables has a specific pathophysiological meaning in the HCM patients’ context [30,31] and a possible capability as an outcome predictor. Particularly, the pVO_2_ is a multidimensional parameter dependent on cardiac output (heart rate * stroke volume) and artero-venous O_2_ extraction [24,32] and it has been extensively shown to be a strong predictor of poor outcome in HCM [14,15,16,17,18] as well as in HF patients [33,34]. The CP%, according to its formula [23,24], magnifies the prognostic power of the pVO_2_ through the ABPRE [17,18] which, in turn, depends on the intrinsic myocardial function/geometry as well on abnormal peripheral autonomic reflexes [35,36]. Eventually, the ventilatory efficiency (i.e., VE/VCO_2_ slope) has been shown to correlate with pulmonary capillary wedge pressure and left ventricular diastolic properties in HCM [37]. Although usually preserved in HCM patients, the VE/VCO_2_ slope tends to worsen significantly only in the late systolic dysfunction phase but it is conceivable that it could mirror also an early exercise-induced left ventricular diastolic functional derangement [16,31,37,38,39].

## 5. Limitations

The relatively small number of patients enrolled, together with the low number of hard events, represents a certain limitation that does not allow us to define the true weight of the genetic analysis results in terms of HCM risk prediction when compared to a full clinical assessment. However, it should be noted that our data do not argue against the overall importance of genetic testing in HCM management. Indeed, the identification of unaffected mutated relatives can be possible only after the detection of P/LP variants in affected HCM probands by gene sequencing [40]. Furthermore, molecular analysis of HCM genes remains one of the pivotal approaches in distinguishing the so-called HCM phenocopies where early diagnosis is crucial to managing it optimally [2,3]. On the other hand, whenever a P/LP variant is not found in HCM patients, the HCM diagnosis should be carefully re-evaluated. Again, concerning risk stratification, a comprehensive clinical assessment might be essential.

Another limitation that needs to be acknowledged is that, apart from the main four genes (*MYBPC3*, *MYH7*, *TNNI3*, *TNNT2*) that were tested in all patients, other genes known to be HCM-related (i.e., *ACTC1, TPM1, TNNC1*, or *MYL2* and *MYL3*) were tested in most but not all patients. However, it should be remarked that these variants are rarely detected in HCM patients [6,7]. Moreover, given the few numbers of patients carrying variants in thin filaments genes (<15%) as well patients with multiple variants (6%) in our study sample, our survival analysis considered the overall impact of the LP/P sarcomeric variants without distinguishing the type of compromised filaments or a possible “gene dosage effect”. Similarly, growing data report possible relationships between specific variants location in functional domains of sarcomeric proteins and prognostic implications. Particularly, pathogenic missense variants located in the converter domain of the *MYH7* gene were found associated with a worse outcome [41,42]. Furthermore, Garcia-Giustiniani and colleagues found a significant association between p.(Gly716Arg), p.(Arg719Trp) and p.(Arg719Gln) variants with a high risk of events (i.e., 50-year survival of only 20% of carriers) [43]. In the present study cohort, although six different pathogenic missense variants falling within the converter domain (p.(Gly716Arg), p.(Arg719Trp), p.(Arg719Gln), p.(Ile736Thr), p.(Gly741Arg) and p.(Arg723Cys)) were found, the small number of patients carrying these variants as well as their relatively young age unable us to support their specific prognostic power. For the same underlying reasons (i.e., small number of patients carrying VUS only), we cannot speculate about an intermediate prognosis in this subset of patients as hypothesized in the SHARE study [13]. Eventually, we found multiple variants in the same gene in 10 patients and in this specific setting, we cannot determinate whether the genetic alterations were on the same chromosome (heterozygous state) or not (compound heterozygous state).

Finally, besides the genetic profile per se, we examined the prognostic effect of several clinical and instrumental variables at a single time point. Accordingly, we cannot exclude that changes in some variables, as for instance an upgrading of treatment during follow-up or upcoming risk factors, altered our survival analysis. However, it is reasonable that seriate clinical and functional evaluations in HCM patients at the highest risk could further magnify our findings rather than rebut them.

## 6. Conclusions

Our data underline the importance of a multidimensional clinical assessment over the genetic testing analysis in the HCM risk stratification. This unexpected finding, different from other literature reports, might be likely explained by the limited size of the sample cohort. The LP/P variants were anyway associated with a more aggressive HCM phenotype in terms of early disease onset, high burden of historical risk factor and, for the first time, poor functional status. Of note, within a number of clinical and instrumental variables, the present study reaffirms the pivotal role of the variables derived from a CPET assessment.

## Figures and Tables

**Figure 1 jcm-09-01636-f001:**
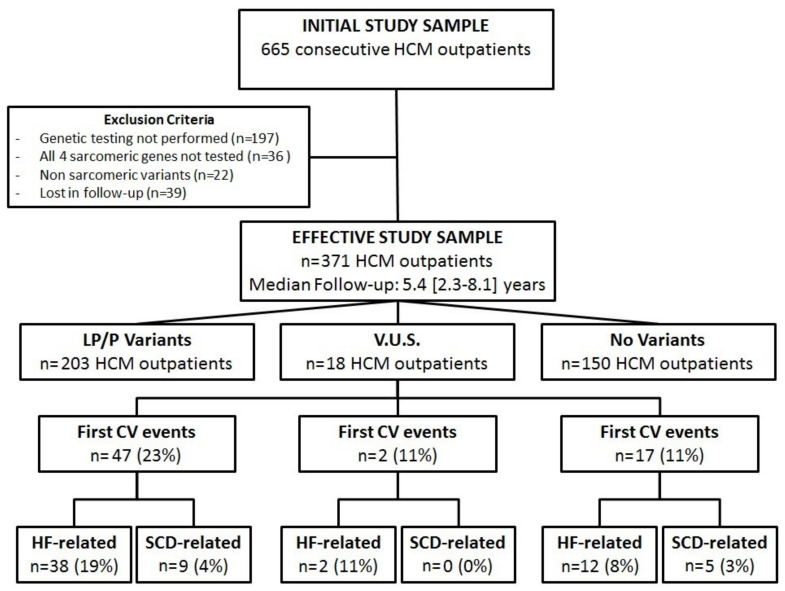
Diagram showing the step-by-step classification of the hypertrophic cardiomyopathy (HCM) population considered in our study. LP/P: likely pathogenic/pathogenic sarcomeric variants; V.U.S.: variant of uncertain significance; CV: cardiac events; HF: heart failure; SCD: sudden cardiac death.

**Figure 2 jcm-09-01636-f002:**
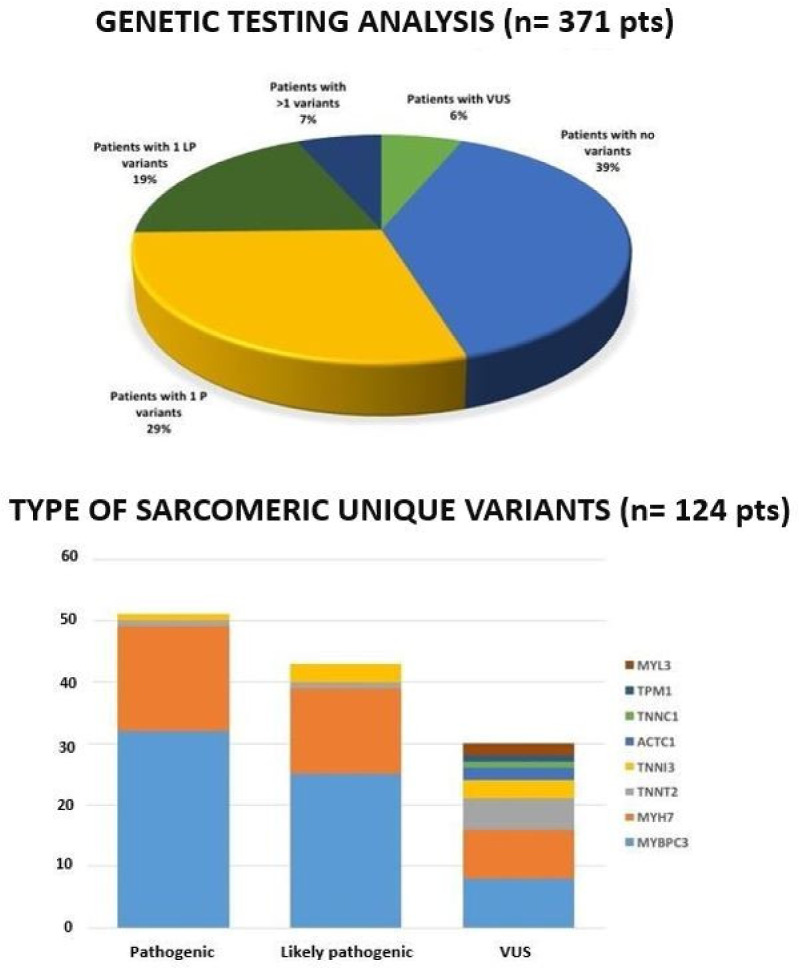
(**Top panel**): results of genetic testing analysis in the overall study sample. (**Bottom panel**): type and distribution of likely pathogenic/pathogenic (LP/P) variants and variants of uncertain significance (VUS). *MYL3*: myosin light chain 3; *TPM1*: tropomyosin 1; *TNNC1*: troponin C1; *ACTC1*: actin alpha cardiac muscle 1; *TNNI3*: cardiac troponin I; *TNNT2*: cardiac troponin T; *MYH7*: β-myosin heavy chain; *MYBPC3*: myosin binding protein C.

**Figure 3 jcm-09-01636-f003:**
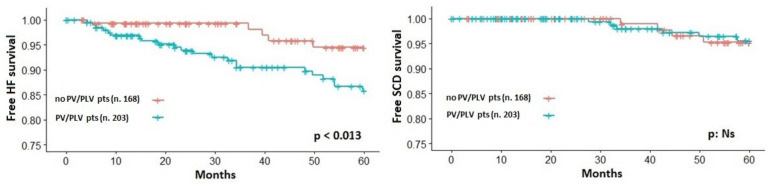
Kaplan–Meier estimator of survival free from heart failure (HF, **left panel**) and sudden cardiac death (SCD, **right panel**) related events according to the presence of likely pathogenic/pathogenic sarcomeric variants (PLV/PV).

**Table 1 jcm-09-01636-t001:** Main clinical variables of the entire study sample at the study run-in (*n* = 371 patients).

General Data	
Age, years	49 ± 16
Male, *n* (%)	238 (64)
Age at diagnosis, years	40 ± 19
NYHA III–IV, *n* (%)	24 (6)
ICD, *n* (%)	43 (12)
Previous myectomy, *n* (%)	20 (5)
**SCD risk factors**	
NSVT, *n* (%)	121 (32)
FH-SCD, *n* (%)	48 (13)
MWT > 30 mm, *n* (%)	26 (7)
Unexplained syncope, *n* (%)	56 (15)
ABPRE, *n* (%)	45 (16)
**Echocardiographic data**	
LVEDd, mm	45 ± 5
LAd, mm	43 ± 7
MWT, mm	20 ± 5
LVOT obstruction, *n* (%)	125 (33)
LVOTG_max_, mmHg	11 (6–39)
LVEF, %	62 ± 7
**CPET data**	
Peak HR, % of predicted	79 ± 13
Peak VO_2_, mL/kg/min	23 ± 7
Peak VO_2_, % of predicted	77 ± 18
Peak SBP, mmHg	164 ± 27
CP%, % of predicted * mmHg	12,937 ± 4300
VE/VCO_2_ slope	28.4 ± 5.5
**Medical treatment**	
β-blocker, *n* (%)	228 (61)
Nondihydropyridine CCB, *n* (%)	31 (8)
ACE-I/ARB, *n* (%)	107 (29)
Diuretics, *n* (%)	78 (21)
Amiodaron, *n* (%)	33 (9)

Data are expressed as mean ± SD, as an absolute number of patients (% on total sample) or as median (25th–75th percentile). VUS: variant of uncertain significance; LP: likely pathogenic; P: pathogenic; NYHA: New York Heart Association; ICD: implantable cardioverter defibrillator; SCD: sudden cardiac death; NSVT: nonsustained ventricular tachycardia; FH: family history; ABPRE: abnormal blood pressure response at exercise; LVEDd: left ventricular end-diastolic diameter; LAd: left atrial diameter; MWT: maximum wall thickness; LVOTG_max_: maximal LV outflow tract gradient; LVEF: LV ejection fraction; SBP: peak systolic blood pressure; HR: heart rate; VO_2_: oxygen uptake; CP: circulatory power; VE/VCO_2_ slope: relation between ventilation versus carbon dioxide production; CCB: calcium channel blocker; ACE-I/ARB: angiotensin-converting enzyme inhibitors/angiotensin receptor blocker.

**Table 2 jcm-09-01636-t002:** Main clinical variables of the study sample at the study run-in according to genetic testing results.

General Data	No Variants and VUS (*n* = 168)	LP/P Variants (*n* = 203)	*p*-Values
Age, years	53 ± 18	45 ± 16	<0.001
Male, *n* (%)	112 (67)	124(61)	NS
Age at diagnosis, years	47 ± 20	35 ± 17	<0.001
NYHA III-IV, *n* (%)	10 (6)	14 (7)	NS
ICD, *n* (%)	11 (6)	32 (16)	0.019
Previous myectomy, *n* (%)	9 (5)	11 (5)	NS
**SCD risk factors**			
NSVT, *n* (%)	48 (28)	73 (36)	NS
FH-SCD, *n* (%)	16 (9)	32 (16)	0.049
MWT > 30 mm, *n* (%)	10 (6)	16 (8)	NS
Unexplained syncope, *n* (%)	29 (17)	26 (13)	NS
ABPRE, *n* (%)	11 (7)	34 (17)	0.003
**Echocardiographic data**			
LVEDd, mm	46 ± 4	45 ± 6	NS
LAd, mm	43 ± 7	43 ± 7	NS
MWT, mm	20 ± 5	20 ± 5	NS
LVOT obstruction, n (%)	79 (47)	45 (22)	<0.001
LVOTG_max_, mmHg	16 (9–39)	10 (5–33)	0.023
LVEF, %	63 ± 4	61 ± 6	<0.001
**CPET data**			
Peak HR, % of predicted	78 ± 12	79 ± 13	NS
Peak VO_2_, mL/kg/min	23 ± 7	23 ± 7	NS
Peak VO_2_, % of predicted	79 ± 18	75 ± 18	0.032
Peak SBP, mmHg	175 ± 26	157 ± 26	<0.001
CP%, % of predicted * mmHg	14,070 ± 4269	12,015 ± 4020	<0.001
VE/VCO_2_ slope	27.5 ± 4.9	29.1 ± 6.0	0.019
**Medical treatment**			
β-blocker, *n* (%)	105 (62)	122 (60)	NS
Non dihydropyridine CCB, *n* (%)	11 (7)	20 (10)	NS
ACE-I/ARB, *n* (%)	58 (34)	48 (24)	0.014
Diuretics, *n* (%)	37 (22)	41 (21)	NS
Amiodaron, *n* (%)	16 (9)	17 (8)	NS

Data are expressed as mean ± SD, as the absolute number of patients (% on total sample) or as median (25th–75th percentile). NS: not significant. See Table 1 for other abbreviations.

**Table 3 jcm-09-01636-t003:** Main significant univariate Cox proportional survival analysis according to the clinical variables for the two main study end-points.

	HF Endpoint (*n* = 52)			SCD Endpoint (*n* = 14)		
	H.R. (95% C.I.)	*p*-Values	C-Index	H.R. (95% C.I.)	*p*-Values	C-Index
Age at CPET	–	NS	–	0.964 (0.934–0.996)	0.038	0.613
Male sex	–	NS	–	–	NS	–
Age at diagnosis	–	NS	–	0.944 (0.906–0.983)	0.006	0.729
FH-SCD	1.869 (1.010–3.460)	0.046	0.522	2.830 (0.892–8.979)	0.077	0.607
Unexplained Syncope	–	NS	–	–	NS	–
NSVT	1.917 (1.102–3.333)	0.021	0.548	–	NS	–
ABPRE	3.418 (1.769–6.605)	<0.001	0.641	–	NS	–
MWT > 30 mm	–	NS	–	3.956 (1.210–12.940)	0.023	0.569
MWT	–	NS	–	1.100 (1.012–1.195)	0.025	0.593
LVOTO	2.110 (1.215–3.664)	<0.01	0.641	–	NS	–
LAd	1.077 (1.039–1.116)	<0.001	0.704	1.054 (0.984–1.129)	0.112	0.660
LVOTG_max_	1.016 (1.008–1.024)	<0.001	0.672	0.971 (0.937–1.007)	0.123	0.577
LVEF	0.929 (0.899–0.959)	<0.001	0.587	–	NS	–
pVO_2_, mL/kg/min	0.851 (0.799–0.905)	<0.001	0.739	–	NS	–
pVO_2_, % of predicted	0.851 (0.799–0.905)	<0.001	0.749	–	NS	–
VE/VCO_2_ slope	1.017 (1.069–1.146)	<0.001	0.724	–	NS	–
CP%	0.998 (0.997–0.999)	<0.001	0.778	0.998 (0.997–1.000)	0.052	0.705
LP or P variants	2.395 (1.171–4.856)	0.013	0.609	–	NS	–

H.R.: hazard ratio; C.I.: confidence interval. See Table 1 for other abbreviations.

**Table 4 jcm-09-01636-t004:** Significant multivariate Cox proportional survival analysis and test for proportional hazards assumption for the two study end-points.

	Multivariate Cox Proportional Survival Analysis					
	HF Endpoint			SCD Endpoint		
	H.R. (95% C.I.)	*p*-Values	C-Index	H.R. (95% C.I.)	*p*-Values	C-Index
**LAd**	1.083 (1.039–1.130)	<0.001	0.839	1.078 (1.005–1.163)	0.0485	0.738
**CP%**	0.998 (0.997–0.999)	<0.001		0.998 (0.9996–1.000)	0.0488	
**VE/VCO_2_ slope**	1.044 (0.999–1.090)	0.05				

See Table 1 and Table 2 for other abbreviations.

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
