# Peer review of "Risk Stratification in Hypertrophic Cardiomyopathy. Insights from Genetic Analysis and Cardiopulmonary Exercise Testing"

_jcm, 2020, doi:10.3390/jcm9061636_

Round 1

Reviewer 1 Report

The authors are trying to test the role of genetic testing over the clinical and functional variables in the hypertrophic cardiomyopathy risk stratification. A retrospective genotype-phenotype correlation was performed to analyze possible differences between patients with and without likely pathogenic/pathogenic variants. The authors did some thorough testing and analysis, although the sample size is a little small, which might affect the conclusion. There are some minor questions that need to be addressed.

  1. The variant of uncertain significance (VUS) is not clearly defined in the manuscript.
  2. The age of the LP/P Variants is much younger than the VUS and no variants, suggesting that the LP/P variants cause an early manifestation of the symptoms. The authors need to consider the age when they draw their conclusions.
  3. Some sentences are very long and might cause confusion, E.g. The LP/P variant was associated with a more aggressive HCM-phenotype but, at multivariate analysis, a significant effect for the HF end-point was found for left atrial diameter (LAd), circulatory-power (peak oxygen uptake*peak systolic blood pressure, CP%) and ventilatory efficiency (C-index=0.839) while only LAd and CP% were associated to the SCD end-point (C-index=0.738).

Author Response

The authors are trying to test the role of genetic testing over the clinical and functional variables in the hypertrophic cardiomyopathy risk stratification. A retrospective genotype-phenotype correlation was performed to analyze possible differences between patients with and without likely pathogenic/pathogenic variants. The authors did some thorough testing and analysis, although the sample size is a little small, which might affect the conclusion.

R. We thank the Reviewer for the great care in reviewing our manuscript and for her/his useful comments. We have done our best to comply with all suggestions. However, should we have failed to correctly/exhaustively address some points, we apologize, and remain open to suggestions for further changes. Obviously, we tried to correct each of our typing errors and to reword some convoluted/misleading sentences. We really apologize and we wish to thank You also for this part of the revision. Thank you again for your insightful comments. We believe that they improved clarity and impact of the present paper.

1. The variant of uncertain significance (VUS) is not clearly defined in the manuscript.

 R. The VUS are clearly defined in the Genetic Testing (paragraph 2.3).

2. The age of the LP/P Variants is much younger than the VUS and no variants, suggesting that LP/P variants cause an early manifestation of the symptoms. The authors need to consider the age when they draw their conclusions.

R. We perfectly agree with You. Indeed we wrote specifically: a) Abstract “[…] The LP/P Variant is associated with a more aggressive phenotype […]”; in the Results “[…] The LP/P variants group showed a younger age, a slightly higher prevalence of FH-SCD and ABPRE and a worse functional capacity in terms of pVO2, CP% and VE/VCO2 slope. With respect to the other clinical features, at the study run-in, the LP/P variants group had a greater prevalence of patients with end-stage phase […]”; in the Discussion: “[…] that HCM patients with LP/P sarcomeric variants tend to present a more aggressive form of disease with an earlier onset, a worse functional status and a greater risk of HF development and HF-related complications, compared to HCM patients with VUS or any variants […]”; in the Conclusions “[…] the presence of LP/P variants has been found to be associated with a more aggressive HCM phenotype in terms of early disease-onset, high burden of historical risk factor and, for the first time, poor functional status […]”

 3. Some sentences are very long and might cause confusion, E.g. The LP/P variant was associated with a more aggressive HCM-phenotype but, at multivariate analysis, a significant effect for the HF end-point was found for left atrial diameter (LAd), circulatory-power (peak oxygen uptake*peak systolic blood pressure, CP%) and ventilatory efficiency (C-index=0.839) while only LAd and CP% were associated to the SCD end-point (C-index=0.738).

R. We really apologize. We checked again the text for possible typos and/or convolute sentences and we tried to improve them. Specifically, in the Abstract, we rephrased as follows: “[…] The LP/P variant was associated with a more aggressive HCM-phenotype. However, left atrial diameter (LAd), circulatory-power (peak oxygen uptake*peak systolic blood pressure, CP%) and ventilatory efficiency (C-index=0.839) were the only independent predictors of HF and just LAd and CP% of the SCD end-point (C-index=0.738) […]”

Reviewer 2 Report

The authors aim to describe the role of genetic testing over the clinical and functional variables, including data from CPET, in patients with HCM.

Although the authors made a significant effort in order to do this, several limitations, particularly regarding the genetic evaluation, should be highlighted. 

1.- First of all, the fact that different genetic sequencing methods are mixed in the analysis is a major limitation (i.e. Sanger vs NGS), as well as the different number of genes evaluated in these patients. Trying to mix such different groups into one, could be a source of major bias. 

2.- Another limitation is that all pathogenic variants are considered equal in this study. Although this has been the traditional approach, recent evidence suggests that specific regions could be associated with worse outcomes, particularly in terms of heart failure and SCD. In this sense, Supplemental Table 2, shows that some patients carry pathogenic missense variants in the converter region of MYH7. Pathogenic variants in this region, particularly those located in the helix region have been consistently reported with adverse events. Therefore, suggesting these variants have the same prognosis as truncating variants in MYBPC3 may not reflect the reality. Unfortunately, the small number of patients with positive genetic study does not allow to make comparisons between different regions of different genes, but this should be taken into consideration. 

3.- How do the authors explain that, in their paper, patients with VUS and no variants have the same prognosis? To date, some papers with a larger number of sequenced patients have suggested that patients with VUS have a worse prognosis when compared to patients without P/LP/VUS variants (and, obviously, better prognosis than those with P/LP). Although this is probably due to the low number of patients, this should be mentioned. Also, adding a Supplementary Table with those VUS could be useful for the scientific community. 

4.- Why were cerebrovascular events (CVE) excluded from the survival analysis?  CVEs are a well-known complication of HCM and the authors should explain this decision. 

5.- The conclusions seem too "radical" considering the aforementioned caveats. I do not consider that their findings "support the clinical assessment over the genetic analysis in the HCM risk stratification". This should be rephrased in order to acknowledge the current limitations. 

I consider that addressing these limitations or at least clearly mentioning them could help increase the value of this paper.

Some minor concerns: 

1.- Although generally well-written, some syntax errors are identified throughout the manuscript.

2.- The references should be re-checked. For example, references 16 and 17 are actually the same reference. Moreover, on page 3, line 137, the authors state that they classified the variants according to the ACMG criteria. However, they reference two papers completely unrelated to the ACMG criteria. One of these references is self-cite. This should be avoided unless completely necessary. In this case, the only acceptable reference is the one belonging to the AMCG guidelines. 

Author Response

The authors aim to describe the role of genetic testing over the clinical and functional variables, including data from CPET, in patients with HCM. Although the authors made a significant effort in order to do this, several limitations, particularly regarding the genetic evaluation, should be highlighted. 

R. We thank the Reviewer for the great care in reviewing our manuscript and for her/his useful comments. We have done our best to comply with all suggestions. However, should we have failed to correctly/exhaustively address some points, we apologize, and remain open to suggestions for further changes. Obviously, we tried to correct each of our typing errors and to reword some convoluted/misleading sentences. We really apologize and we wish to thank You also for this part of the revision. Thank you again for your insightful comments. We believe that they improved clarity and impact of the present paper.

1.- First of all, the fact that different genetic sequencing methods are mixed in the analysis is a major limitation (i.e. Sanger vs NGS), as well as the different number of genes evaluated in these patients. Trying to mix such different groups into one, could be a source of major bias. 

R. The Reviewer correctly underscores the issue of the different genetic sequencing methods in the different centers/period. Being aware of this limitation, we rephrased the paragraph 2.3 as follows: “[…]The genes analyzed in the participating laboratories from the HCM centres have changed and/or added over time. In this retrospective study we report the genotype-phenotype correlation analysis only for those patients screened at least for the sarcomeric genes MYBPC3 (myosin binding protein C, cardiac) MYH7 (β-myosin heavy chain, cardiac), TNNI3 (troponin I, cardiac) and TNNT2 (troponin T type 2, cardiac). All coding regions and boundaries of flanking introns ± 25 were analyzed for all genes considered in this study. In addition, all variants reported in this paper, identified by NGS technology, were validated by Sanger sequencing. Patients with variants located in non-sarcomeric genes were also excluded. […]”. We hope that the Reviewer is satisfied with our change.

2.- Another limitation is that all pathogenic variants are considered equal in this study. Although this has been the traditional approach, recent evidence suggests that specific regions could be associated with worse outcomes, particularly in terms of heart failure and SCD. In this sense, Supplemental Table 2, shows that some patients carry pathogenic missense variants in the converter region of MYH7. Pathogenic variants in this region, particularly those located in the helix region have been consistently reported with adverse events. Therefore, suggesting these variants have the same prognosis as truncating variants in MYBPC3 may not reflect the reality. Unfortunately, the small number of patients with positive genetic study does not allow to make comparisons between different regions of different genes, but this should be taken into consideration. 

R. We perfectly agree with the Reviewer’s concern but, as he/she correctly underlined, we cannot support any thesis about this topic because of the small sample. However, we highlighted extensively the issue in the Limitation paragraph as follows: “[…] growing data report possible relationships between specific variants location in functional domains of sarcomeric proteins and prognostic implications. Particularly, pathogenic missense variants located in the converter domain of the MYH7 gene, have been found associated to a worse outcome [41,42]. Furthermore, Garcia-Giustiniani and colleagues found a significant association between p.(Gly716Arg), p.(Arg719Trp) and p.(Arg719Gln) variants with a high risk of events (i.e. 50-years survival of only 20% of carriers) [43]. In the present study cohort, although six different pathogenic missense variants falling within the converter domain [p.(Gly716Arg), p.(Arg719Trp), p (Arg719Gln), p.(Ile736Thr), p.(Gly741Arg) and p.(Arg723Cys)] have been found, the small number of patients carrying these variants as well as their relatively young age unable us to support their specific prognostic power […]”. As You can note, we also added three references dealing with this topic (i.e. ref. n.41-43)

3.- How do the authors explain that, in their paper, patients with VUS and no variants have the same prognosis? To date, some papers with a larger number of sequenced patients have suggested that patients with VUS have a worse prognosis when compared to patients without P/LP/VUS variants (and, obviously, better prognosis than those with P/LP). Although this is probably due to the low number of patients, this should be mentioned. Also, adding a Supplementary Table with those VUS could be useful for the scientific community. 

R. As per point 2, it is possible that the lack of significance is due to the small number of our study cohort (just 23 patients carrying a VUS only on a total of 371 patients). Thus, we added the following short sentence in the Limitation section “[…] For the same underlying reasons (i.e. small number of patients carrying VUS only), we cannot speculate about an intermediate prognosis in this subset of patients as hypothesized in the SHARE study […]”. Conversely, concerning the Table with VUS, it is likely that there was a misunderstanding or a problem with the submission procedure because we supplied it also in the first version (Table 2 S. List of variants of uncertain significance (VUS) identified).

4.- Why were cerebrovascular events (CVE) excluded from the survival analysis?  CVEs are a well-known complication of HCM and the authors should explain this decision. 

R. Honestly, due to the small number of cerebrovascular events (2 fatal strokes and 5 non-fatal events which, in some cases, were recorded in patients who reached also the other pre-specified end-points), we preferred to analyze directly two “cardiac” end-points avoiding a possible text/statistical overload linked to an adjunctive composite end-point. However, we now explain the reasons underlying the abovementioned approach in the revised Methods by writing “[…]To avoid a composite cardiovascular end-point including also cerebrovascular events, due to the small number of such events and in accordance with other studies by our research group [17,18,22], death due to ischemic or hemorrhagic stroke (n. 2 events) and non-fatal cerebrovascular (n. 5 events) as well as death due to non-cardiovascular causes (n. 2 events) were excluded from the survival analysis. […]”.

5.- The conclusions seem too "radical" considering the aforementioned caveats. I do not consider that their findings "support the clinical assessment over the genetic analysis in the HCM risk stratification". This should be rephrased in order to acknowledge the current limitations. I consider that addressing these limitations or at least clearly mentioning them could help increase the value of this paper.

R. Again, thank you for Your insightful comments. We modified some sentences in order to tune down our conclusion about the role of genetic testing in the HCM risk stratification. Specifically, we introduced the following changes: the new title of the article is “Risk Stratification In Hypertrophic Cardiomyopathy. Insights From Genetic Analysis and Cardiopulmonary Exercise Testing”; in the Limitation “[…]limitation that does not allow us to define the true weight of the genetic analysis […]”; eventually, in the Conclusions section “[…]Our data underline the importance of a multidimensional clinical assessment over the genetic testing analysis in the HCM risk stratification. This unexpected finding, different from other literature reports, might be likely explained by the limited size of the sample cohort. The LP/P variants were anyway associated with a more aggressive HCM phenotype in terms of early disease-onset, high burden of historical risk factor and, for the first time, poor functional status. Of note, within a number of clinical and instrumental variables, the present study reaffirms the pivotal role of the variables derived from a CPET assessment […]”.

Some minor concerns: 

1.- Although generally well-written, some syntax errors are identified throughout the manuscript.

R. We really apologize. We checked again the text for possible typos and/or convolute sentences and we improved them.

2.- The references should be re-checked. For example, references 16 and 17 are actually the same reference. Moreover, on page 3, line 137, the authors state that they classified the variants according to the ACMG criteria. However, they reference two papers completely unrelated to the ACMG criteria. One of these references is self-cite. This should be avoided unless completely necessary. In this case, the only acceptable reference is the one belonging to the AMCG guidelines. 

R. Thank you very much for noticing the mistakes! Accordingly we modified the references as suggested.

Round 2

Reviewer 2 Report

The authors have adequately addressed the previously mentioned issues or, when not possible, added approriate sentences to the text explaining the limitations. 

I consider the manuscript is suitable for publication in its present form.